# Forest Tourism and Recreation Management in the Polish Bieszczady Mountains in the Opinion of Tourist Guides

**Emilia Janeczko [1,\*], Joanna Pniewska [1,\*] and Ernest Bielinis [2]** 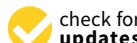

[1]  Department of Forest Utilization, Institute of Forest Sciences, Warsaw University of Life Sciences, Nowoursynowska 159, 02-776 Warsaw, Poland

[2]  Department of Forestry and Forest Ecology, Faculty of Environmental Management and Agriculture, University of Warmia and Mazury in Olsztyn, Pl. Łódzki 2, 10-727 Olsztyn, Poland; ernest.bielinis@uwm.edu.pl

\*  Correspondence: emilia.janeczko@wl.sggw.pl (E.J.); joanna_pniewska@mail.sggw.pl (J.P.)

**Abstract:** This article presents results of research carried out in 2018 that aimed to determine the opinions of Bieszczady mountain guides on the scope of development of tourism and recreational infrastructure in the Bieszczady forests, Poland. The online survey included questions regarding nature protection in the Bieszczady region, factors limiting opportunities for tourism and recreation development in the Bieszczady forests, and the needs regarding new elements of tourism and recreation management of the area. Our research results indicate that the greatest impediments to the recreational use of the forest result from temporary restrictions on forest access, which are related to hunting or forest-management works. Most the interviewed guides were against further development of the tourist and recreational infrastructure in the Bieszczady forests. They were also in favor of extending the nature protection area in Bieszczady. Statistical analyses using the Kruskal–Wallis test showed that persons who are against, in favor of and neutral on extending the nature protection area in the Bieszczady forests varied significantly in their views on issues such as hunting or restrictions on forest access related to forest-management works. Compared to the other respondents, the supporters of extending the range of protected areas were more frequently against designating new recreation spaces or bonfire places in the Bieszczady forests.

**Keywords:** touristic infrastructure; forest recreation management; environmental protection; questionnaires; Carpathian Mountains

## 1. Introduction

For the past several decades, mountain tourism has developed significantly across the world [1]. At the same time, interest in environment protection has been increasing, and ecotourism has become increasing popular [2–5]. In a broader approach, mountain tourism covers tourist activities that take place in the mountain environment, with a close connection to nature and the culture of a specific region [6]. A mountain area can be characterized by a vast diversity of tourism resources that create good conditions for winter sports and other sport and recreational activities, such as trekking, mountain running, cycling or horse riding. Mountain forests—along with their specific flora and fauna—are also a basis for the development of various forms of tourism and recreation (ecotourism, terrain therapy, hunting, fishing, single tracking). Forest recreation is a specific form of an activity in the open air that supports physical activity and psychological wellbeing [7]. Non-consumptive wildlife recreation such as wildlife watching, wildlife photography and bird watching and bird feeding, is a popular activity for many people and provides a significant economic contribution to

the local economy. Expenditures from wildlife watching activities generate employment and income in various manufacturing industries and service sectors [8]. Hunting activities and wildlife-related recreational services have also been increasingly recognized for the benefits they can provide to the local economies [9]. In many developing countries, mountain forests play a significant cultural and spiritual role, thus they quite often obtain the status of protected areas [10].

Mountain ecosystems are especially sensitive, and their beauty and quality subject to the pressure of the urbanization process, mining activities, changes in the agriculture and forest management, but also to the strong influence of tourism and recreation [11]. Intensive tourism development has provoked international discussions about mountain areas and their sustainable development [12]. Examples of essential documents promoting an integrated approach to the nature and cultural mountain heritage include the Framework Convention on the Protection and Sustainable Development of the Carpathians from 2003 [13], which was ratified by Poland in 2006 or the Protocol on the Sustainable Tourism to the aforementioned convention, whose provisions entered into force on 29 April 2013 [14]. The main idea of this document was to coordinate a transboundary cooperation in terms of, among others, strengthening a role of tourism in the sustainable forest management and the sustainable development of a tourism infrastructure in the Carpathians. Forest recreational facilities and improved access to the forest are considered top development priorities and efficient management of the mountain tourism [15], because the tourism infrastructure constitutes a great advantage of each region, compared to other regions and helps them to succeed in the tourism development [16]. The sustainable development of tourism and recreation in the mountain forest requires a specific knowledge about experiences and behaviors of visitors and their consequences, as well as how the recreation development should be used [10].

In the past years, a number of studies have been conducted with the aim was to identify social expectations towards tourism and recreation in forests [17,18]. Thus, demands and recommendations resulting from those research allows adopting a model of tourism and recreation development which would support environment protection, while at the same time, meet needs and preferences of different groups of users [19]. A thorough understanding of visitor preferences about resources (e.g., infrastructure resources), social conditions (e.g., behavior of visitors) and management conditions (e.g., forest economy) of the natural environment is crucial when developing an efficient strategy for the landscape management [17,20]. Koemle and Morawetz [21] considered that managing recreational areas in the open air requires a balance between interests of many groups of users who perform recreational activity and users who use resources professionally (e.g., hunters or foresters). Tourist guides form an important group of interest for the tourism development in mountain forests. In accordance with Polish law [22] tourist guides are "people who professionally guide tourists or visitors in selected areas, towns and facilities, provide subject-matter information and take care of tourists and visitors within the scope that results from the agreement between them". Guides are a specific group of people who know the area and its history thoroughly, cooperate with tourists and monitor their outdoor behavior.

Mountain tourist guides play a significant role in the environmental education and, at the same time, have a special responsibility during their trips [23]. By entering the field, enabling a direct contact with nature, promoting its beauty and drawing attention to its destruction, they create a unique atmosphere that allows them to influence both knowledge and emotions of their groups. Guided tours are undoubtedly a form of educational field activities (both in formal and informal education). According to Dąbrowski [23], the status of a tourist guide and the nature of their relationship with a group give basis for highly effective environmental education. These include, in particular, such factors as authority, impressiveness, trust, informal character of relations with a group, etc. However, such positions may pose a risk of negative effect, even anti-education, if the tourist guide represents a wrong attitude in this respect. This issue becomes particularly important when there are protected or sensitive areas along the tour route, such as tree logging sites, etc. When tourist guides share their opinions concerning forests condition, actions undertaken to develop the tourism and recreation infrastructure in the forests, they have a significant impact on the public reception of the forest economy.

Currently in Poland, after the case of the Białowieża Forest, the greatest source of conflict between foresters and the public refers to the Eastern Carpathians, which are called "the Carpathian Forest" by various environmental organizations. Public attention is now focused on Beskidy. The subject of this conflict primarily refers to the size of logging works in the forest, as well as the extension of the communication network to make the forest more accessible. Ecologists believe that the forests of Bieszczady should be protected unconditionally. On their initiative, over 140,000 signatures were collected under the petition to create another national park in Polish mountains—the Turnicki National Park [24]. According to the ecologists, tree-felling devastates the landscape and pose a threat to the existence of many valuable plant species [25]. Considering the tourism development, forest harvesting works also lead to temporary restrictions on land availability. Ecological organizations also point out that the intensive expansion and modernization of forest roads and skidding trails, carried out in the recent years in the Bieszczady Mountains, not only has a significant impact on the environment (e.g., fragmentation and interruption of the continuity of the forest complex), but also leads to the creation of grossly disharmonious elements of the forest landscape [26]. Dąbrowski [23] points out that the attractiveness of Polish mountains is systematically falling. The mountain landscape is becoming unappealing, which is caused by increasingly aggressive landscape accents. Its natural area is also shrinking due to the construction of new buildings, roads and cable cars reaching higher zones of the mountains. According to foresters, the volume of wood harvesting is dictated by breeding efforts—including the need to uncover natural regeneration—which is in accordance with current legal regulations. On the other hand, the extension and modernization of roads is an effect of correcting errors and previous omissions in this area, as it favors the improvement of fire safety and, at the same time, allows rational control of tourist traffic in order to relieve the most valuable nature areas of the Bieszczady Mountains [27].

It follows from the above that the main obstacles to the implementation of tourism and recreation in the Bieszczady forest include limitations in the availability of land due to harvesting, breeding and protection works as well as disturbances in the landscape esthetics (e.g., introduction of landscape alien elements and logging), as well as problems concerning the functioning of the tourism infrastructure (e.g., underdeveloped network of trails, poor condition of forest infrastructure). The Bieszczady forests are perceived as the most beautiful and virgin in Poland. For this reason, for years they have been eagerly visited by hunters—both from Poland and abroad—hunting for big animals. There is no statutory restriction on entering the forest during group hunting in Poland. Nevertheless, where hunting takes place, individual forest districts may introduce a temporary ban on entry, which means that it is very difficult for all other people who rest in the forests, especially in the period between 1 October and 31 January, when the largest number of hunting takes place [28,29].

Thus, the aim of this research is to identify opinions of the Beskidy mountain guides on possibilities of the tourism and recreation development in the Bieszczady forests, as well as the scope of nature protection measures in this area.

Our research has both cognitive and practical implications. It allows filling gaps in our knowledge about social expectations concerning nature protection in the Bieszczady Mountains, as well as learning the opinion of tourist guides about the activity of foresters in this area. In practical terms, the results of our research are useful for creating programs and policies for the development of tourism and recreation in the Bieszczady Mountains. These results provide support for wildlife managers and policy makers at both national and regional level who may be interested in conducting more effective environmental education and optimizing the scope of recreational management of the Bieszczady forests. This is highly significant because the development of tourism and recreational infrastructure in the Bieszczady forests can lead to a significant economic contribution in this relatively poor region of Poland.

## 2. Materials and Methods

### 2.1. Study Sites

The Polish Bieszczady covers southeastern part of the country (Figure 1).

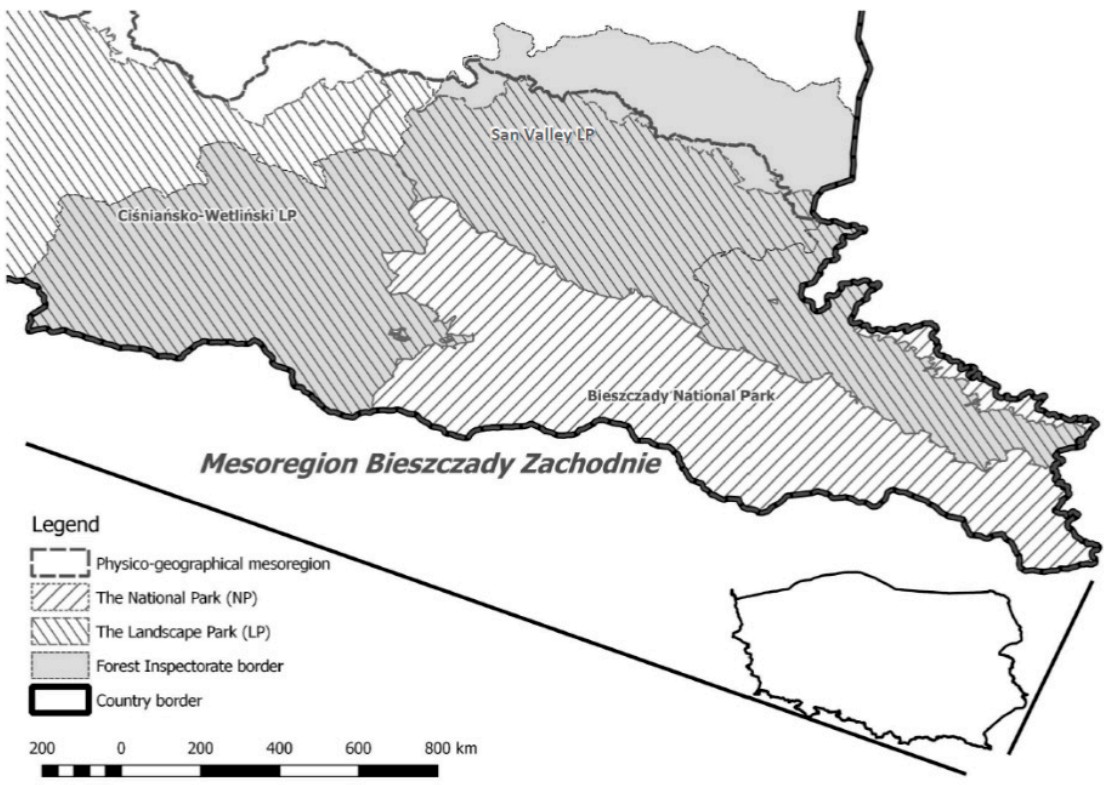

**Figure 1.** Location of Bieszczady (own study).

The Żuków chain constitutes the north border, which separates the Bieszczady Mountains from the Sanok-Turka Mountains. The west and southwest border runs along a train railway from Zagórz to Łupków. In the south, there is a border chain which stretches from Slovakia to Ukraine, which is located between the Uzhok Pass (853 m) to the west and the Łupków Pass (640 m) to the east. Part of the Polish-Ukrainian border also constitutes the east border of the Bieszczady mountains [30]. Mt. Tarnica (1346 m) is the highest summit of this mountain range. The Bieszczady mountains are characterized by a high rate of forestation of 37.79% in the Podkarpacie Province, while in the Bieszczady district, which is a part of this province—even 69.75% [31]. The total area of the forests in the Podkarpacie Province amounts to 683.6 K ha. According to the property structure of forests in the province, as in the rest of Poland, public forests constitute a vast majority—561.0 K ha (among which 532.6 K ha belong to State Treasury and 28.4 K ha belong to commune authorities). Private forests constitute 122.6 K ha. Referring to the forests of State Treasury: 489.0 K ha are managed by the State Forests National Forest Holding; 40.9 K ha by the national parks and the remaining 2.7 K ha belong to other units of the Treasury [32]. The area discussed in this article subjects to the Bieszczady National Park, which was established in 1973. The national park area amounts to 29.2 K ha, including 24.4 K ha covered with forests [33]. The Bieszczady National Park is the biggest national park in the Polish mountains and third biggest national park in the country.

### 2.2. Participants

A total of 130 mountain guides took part in the survey. A detailed breakdown of the participants is as follows: 78 persons were from 4 student associations of Beskidy mountain guides (Lublin, Katowice,

Rzeszów, Warsaw), 40 persons were from 2 student associations of mountain guides of Harnasie and Kraków and 12 persons were from the Tourist Guide Association in Gdańsk, the Student Mountain Guide Club and two other associations. the majority of the participants had a higher education, which resulted from the fact that pursuant to the law [22] a tourist guide is required to have at least secondary education or professional secondary education [art. 21(1)(1b)]. Demographic data of the participants is presented in Table 1.

**Table 1.** Demographic information of study participants.

| **Gender** | Female 40% | Male 60% | | | |
|---|---|---|---|---|---|
| **Age** | <25 9% | 26–35 55% | 36–45 23% | 46–56 5% | >56 8% |
| **Educational level** | High school 6% | University 94% | | | |
| **Educational profile** | Natural science 32% | Technical 28% | Humanities 24% | Economic 8% | Other 8% |
| **Place of residences** | Village 14% | City with up to 200,000 inhabitants 22% | City with over 200,000 inhabitants 64% | | |

All participants were informed about the nature and objective of the research and informed consent was obtained from each of them. The authors ensure that this research was completed without a potential conflict of interests. All procedures carried out in this paper were compliant with the ethical standards of the Polish Science Ethics Committee and the Declaration of Helsinki of 1964, as amended.

*2.3. Research Procedure*

The research was conducted between November 2018 and January 2019 in the form of an online survey. The research material was collected via the online survey in a Google form, published on the official fora of the mountain guides from the aforementioned associations. The questionnaire included demographic questions (sex, age, education, education profile and place of residence) which aimed at defining structure of the respondents, as well as subject-matter questions which aimed at defining respondent opinions on the following topics: nature protection in Bieszczady; factors hindering recreational comfort in the forest (e.g., difficulties caused by temporary restricted access to the forest, difficulties caused by a malfunction of the tourist and recreational infrastructure; and violating spatial order and esthetic values of the space); and the need for new elements of tourist and recreational development in the region. A question about opinion on the nature protection in Bieszczady allowed to categorize each respondent to one of three groups (supporters, opponents and neutral persons towards the expansion of the protected areas). Other questions in the survey concern questions based on the Likert scale, which enabled to obtain answers revealing a level of agreement on a specific opinion. The scale consists of a list of five scaled answers ascending from the level of the strong agreement to the strong disagreement. All answers were anonymous and confidential. Raw data from the interviews were used for statistical analyses. Within the statistical analysis a standard deviation of a proportion (Sp) was estimated by means of the Mann–Whitney U test; this showed whether the median was different in the compared samples. A one-factor analysis of variance was also carried out, using the Kruskal–Wallis test. Answers concerning recreation impediments and infrastructure were analyzed by principal component analysis (PCA) and the analysis of variance (ANOVA) tests, respectively. Statistical analyses were conducted with RStudio Desktop AGPL v3 (RStudio, Boston, MA, USA) and Statistica 13 (TIBCO Software, Inc.).

## 3. Results

The analysis of the research results shows that most Beskidy guides (43%) felt that the area of protected areas in the Bieszczady Mountains should be extended. A significant group of the guides (28 percent) was against expanding the protected areas. Another group of respondents (28 percent) had a neutral opinion on this subject.

According to the Bieszczady mountain guides, the greatest impediment to the recreational use of the forest results from a temporary limited access to the forest which is related to hunting (Table 2).

**Table 2.** Respondent opinions on difficulties in the implementation of tourism and recreation in the Bieszczady forests.

| | | | Strongly Disagree | Rather Disagree | No Answer | Rather Agree | Strongly Agree |
|---|---|---|---|---|---|---|---|
| Access limitations | Protection and silviculture works | % | 7% | 28% | 16% | 33% | 15% |
| | | Sp | 0.0223 | 0.0396 | 0.0323 | 0.0413 | 0.0316 |
| | Hunting | % | 5% | 16% | 12% | 22% | 44% |
| | | Sp | 0.0198 | 0.0323 | 0.0288 | 0.0365 | 0.0435 |
| | Timber harvesting-related works | % | 5% | 22% | 22% | 28% | 24% |
| | | Sp | 0.0198 | 0.0361 | 0.0361 | 0.0392 | 0.0374 |
| Violation of the landscape esthetics | Litter close to trails | % | 2% | 17% | 19% | 34% | 28% |
| | | Sp | 0.0108 | 0.0329 | 0.0346 | 0.0415 | 0.0396 |
| | "Unnatural" landscape in the forest | % | 5% | 20% | 15% | 27% | 32% |
| | | Sp | 0.0198 | 0.0351 | 0.0316 | 0.0389 | 0.0410 |
| | Clearcuttings close to tourist trails | % | 3% | 26% | 12% | 34% | 25% |
| | | Sp | 0.0151 | 0.0385 | 0.0280 | 0.0415 | 0.0382 |
| Functioning of the forest infrastructure | No public consultation on designating tourist trails | % | 2% | 15% | 36% | 32% | 15% |
| | | Sp | 0.0132 | 0.0310 | 0.0421 | 0.0410 | 0.0310 |
| | Poorly developed touristic network | % | 15% | 52% | 15% | 15% | 2% |
| | | Sp | 0.0316 | 0.0438 | 0.0316 | 0.0316 | 0.0108 |
| | Condition of the forest infrastructure | % | 12% | 37% | 22% | 24% | 5% |
| | | Sp | 0.0280 | 0.0423 | 0.0365 | 0.0374 | 0.0198 |

The majority of the respondents (66 percent in total, including 44 percent who said, "strongly agree)" indicated hunting as the major factor limiting recreational activities in the Bieszczady forests. Furthermore, works related to timber harvesting, forest machinery and vehicles contribute to temporary limitations in performing touristic and recreational functions of the forest. Such opinion was shared by 52 percent of the respondents in total.

Taking into account the demographic characteristics of the respondents, it was determined that opinions related to temporary limitations in forest access due to protection and silviculture works differ mainly with respondent age (Figure 2).

The diagram shows the differentiation of the respondent answers due to demographic characteristics, not the direction of the groups' responses. The most similar opinions were held by respondents aged under 25 and over 46. Other factors, such as education level, place of residence and gender, were not significant, but among them the similarity of responses of women and people with university education, as well as men and people with higher education, was noticed. The same relation occurred in the case of people living in cities with more than 200,000 residents and people aged 36–45, as well as residents of villages and towns up to 200,000 and respondents aged 26–35.

PCA analysis also showed that opinions related to hunting as temporary limitation in forest access mainly differ due to respondent age. Respondents under 25 and over 56 years old had a similar opinion (Figure 3). In this analysis similar demographic relations occurred to those in the PCA analysis concerning protection and silviculture works.

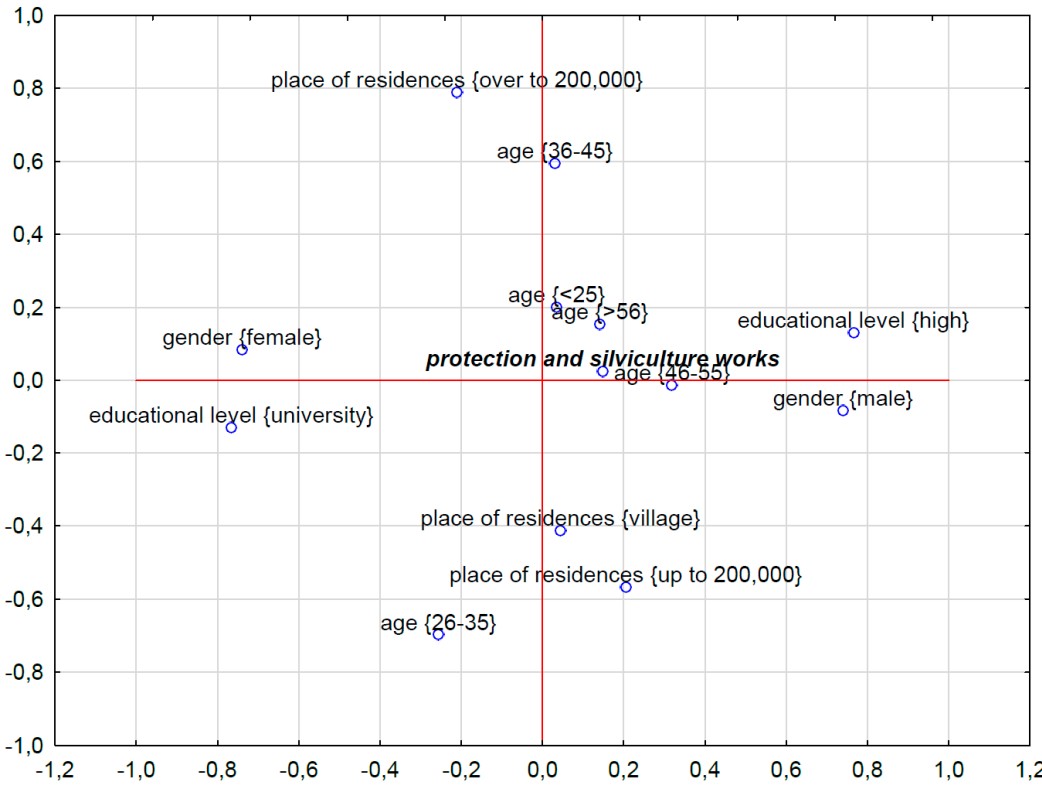

**Figure 2.** Principal component analysis (PCA) analysis of correlation of respondent demographic characteristics and answers on recreation impediment due to protection and silviculture works (own study).

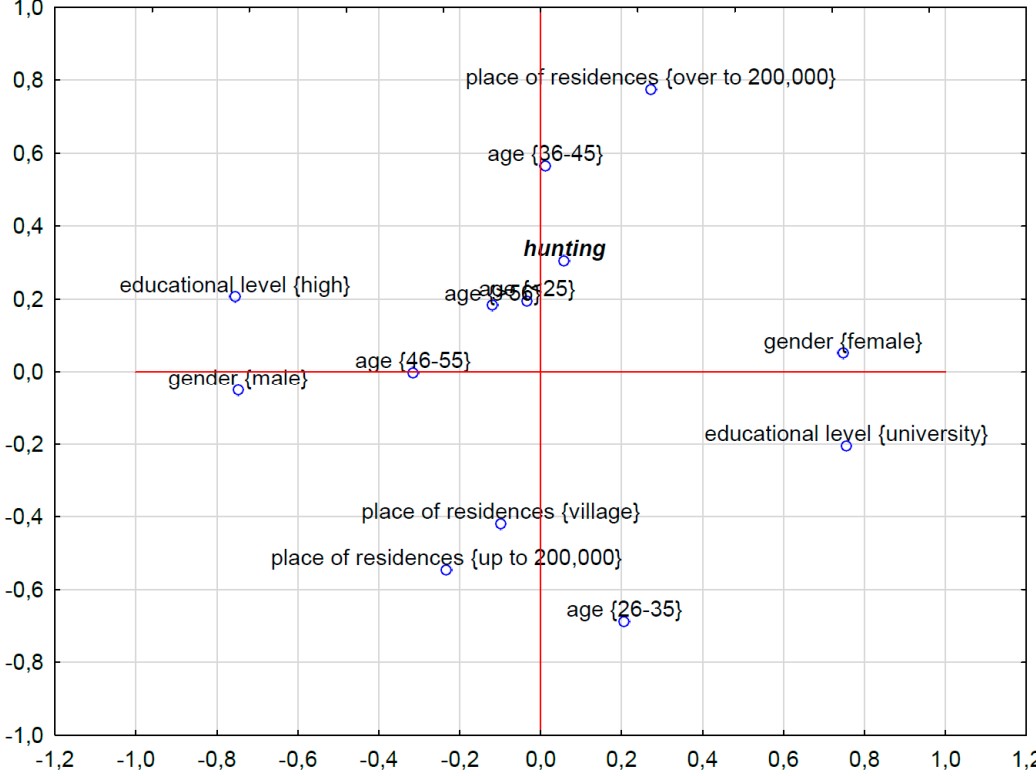

**Figure 3.** PCA analysis of correlation of respondent demographic characteristics and answers on recreation impediment due to hunting (own study).

According to the respondents, a significant factor limiting the comfort of recreation in the forest was defined as a violation of the landscape esthetic. The violation was related to trash visible in close proximity to routes and recreational places, clearcutting performed close to the tourist trails or hard surface and asphalt roads running through the woods which are perceived as an unnatural element in such landscape. Sixty-two percent of the respondents in total were convinced (including 28% who were strongly convinced) that trash significantly reduces comfort of the recreation in the forest. Other problems that significantly limit a possibility of implementing touristic and recreational functions of the forest are related to the functioning of the infrastructure. Respondents said that the poorly developed trail network (67 percent) or bad conditions of the infrastructure (49 percent), i.e., neglected shelters or trees lying across the trails, do not cause problems, while another quite large group of respondents (47 percent in total) indicated that actions aimed at increasing the participation of the society in designating tourist/ recreational routes should be improved.

Using Mann–Whitney test it was established that, statistically, the significant differences between the respondent opinions ($p < 0.05$) can be observed in terms of issues such as "no public consultations at the stage of trails marking" and "poorly development touristic network" (Figure 4).

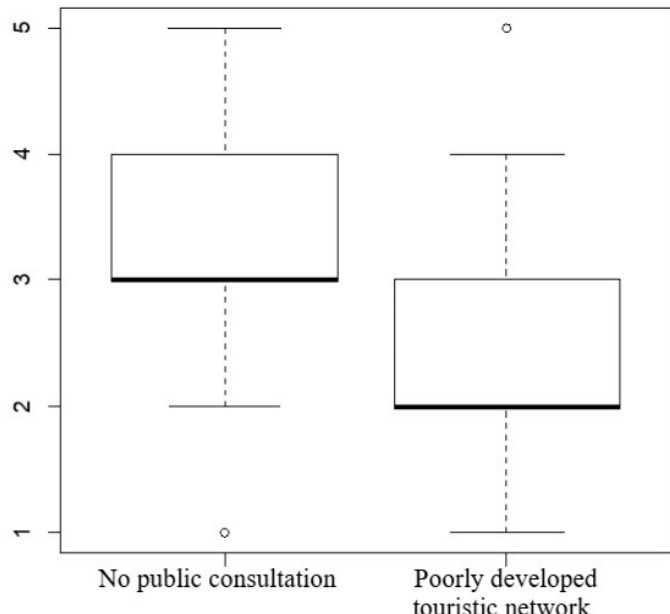

**Figure 4.** Comparison of the average median for selected respondent opinions on factors limiting a possibility of tourism and recreation in the Bieszczady forests.

Statistical analyses showed that there are significant differences between persons who are against, in favor of and neutral on extending the nature protection area of in the Bieszczady forests in relation to the hunting or restricted forest access due to forest-management works. Among three groups of respondents (opponents, supporters and people neutral to the expansion of the protected area), the opponents much less indicated hunting as a factor limiting access to the forest ($p = 0.0297$). In total (the sum of responses "strongly" and "rather" agree) this factor was indicated by 27% of respondents from the group, "opponents of the increased protection area" and 57% of respondents from the group, "supporters of the increased protection" and 41% "undecided/neutral persons" (Figure 5).

The supporters of the extension of the protected area more often than other groups indicated works carried out in the forest as the major factor limiting access to the forest for tourism and recreation ($p = 0.0249$). In total (the sum of responses, "strongly" and "rather" agree) this factor was indicated by 59% of respondents from the group, "supporters of the increased protection", 38% "undecided/neutral persons" and 43% of respondents from the group, "opponents of the increased protection area".

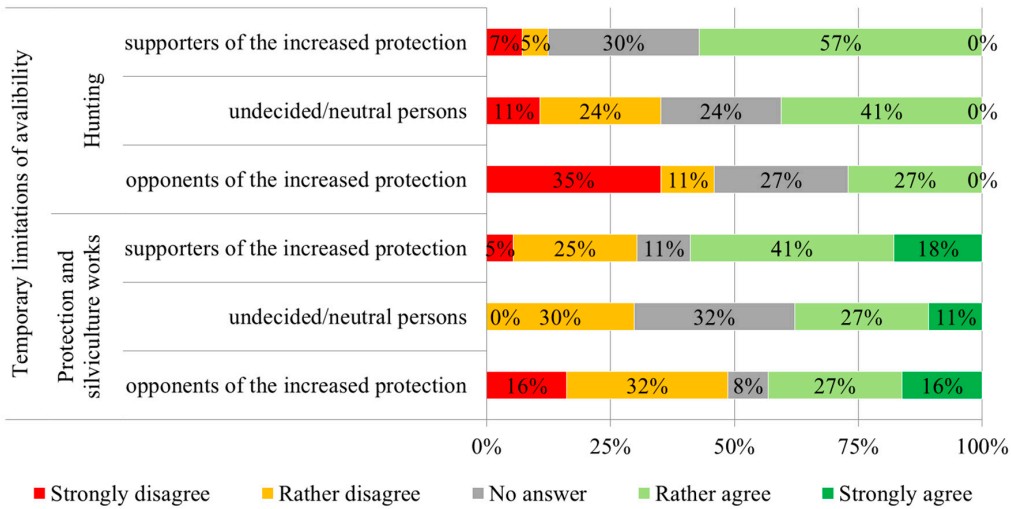

**Figure 5.** Comparison of significantly different (*p* < 0.05) opinions on temporary limitations of the access to the forest, considering breakdown of the respondents into supporters and opponents of nature protection and neutral persons.

Most of the interviewed guides were against any further development of the touristic and recreational infrastructure in the Bieszczady forest. The greatest differences were observed in terms of walking tourist trails. Twenty-eight percent and 48 percent of respondents were strongly against or rather against, respectively, a development of footpaths. Thirty percent of respondents approved marking new cycle routes while 49 percent of them were against it (Table 3). A majority of the respondents was against a creation of new educational trails (53 percent) and further initiatives undertaken in terms of the creation of new cycle routes (49 percent). Among linear elements of the area development equestrian trails were preferred the least (15 percent). Fifty-five percent were against developing equestrian trails (including 20 percent—definitely against). Considering the rest of the (spatial and single) elements of the touristic and recreational infrastructure in the forest, it was noticed that a significant percentage of the respondents (70 percent in total) indicated a need to designate a greater number of campsites and bonfire places in the Bieszczady forests (51 percent).

**Table 3.** Respondent opinions on a demand for touristic and recreational infrastructure in the Bieszczady forests.

| | | | Strongly DISAGREE | Rather DISAGREE | No Answer | Rather AGREE | Strongly AGREE |
|---|---|---|---|---|---|---|---|
| Linear elements | Footpaths | % | 28% | 48% | 7% | 13% | 4% |
| | | Sp | 0.0392 | 0.0438 | 0.0223 | 0.0296 | 0.0169 |
| | Equestrian trails | % | 20% | 35% | 31% | 11% | 4% |
| | | Sp | 0.0351 | 0.0417 | 0.0405 | 0.0272 | 0.0169 |
| | Cycling trails | % | 19% | 30% | 21% | 21% | 9% |
| | | Sp | 0.0346 | 0.0402 | 0.0356 | 0.0356 | 0.0254 |
| | Educational trails | % | 13% | 40% | 17% | 27% | 3% |
| | | Sp | 0.0296 | 0.0430 | 0.0329 | 0.0389 | 0.0151 |
| Other | Bonfire places | % | 11% | 27% | 12% | 29% | 22% |
| | | Sp | 0.0272 | 0.0389 | 0.0280 | 0.0399 | 0.0361 |
| | Recreational places | % | 9% | 32% | 16% | 32% | 11% |
| | | Sp | 0.0254 | 0.0408 | 0.0323 | 0.0410 | 0.0272 |
| | Camping sites | % | 5% | 13% | 12% | 48% | 22% |
| | | Sp | 0.0198 | 0.0296 | 0.0280 | 0.0438 | 0.0365 |
| | Parking in the forest | % | 16% | 26% | 22% | 26% | 10% |
| | | Sp | 0.0323 | 0.0385 | 0.0361 | 0.0385 | 0.0263 |

The Mann–Whitney test showed that, statistically, the significant differences between the respondent opinions ($p < 0.05$) can be observed in terms of issues such as, "scarcity of footpaths" and "necessity to designate more camping sites" (Figure 6).

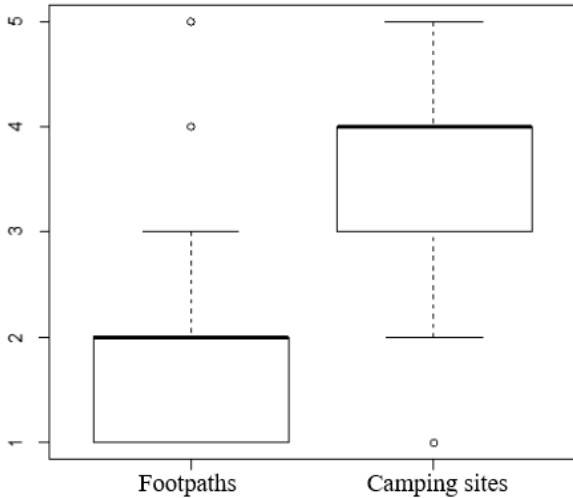

**Figure 6.** Comparison of the average of ranks for selected respondent opinions on a functioning of the recreational infrastructure in the forest.

The statistical analysis conducted with the Kruskal–Wallis test showed a close statistical significance of differences in opinions between the supporters of the extension of spatial forms of the nature protection and their opponents or respondents undecided in the matter of the expansion of recreational infrastructure in forests. The former group had much stronger negative opinions against designating new recreational sites ($p = 0.0610$) or bonfire places ($p = 0.0633$)—Figure 7.

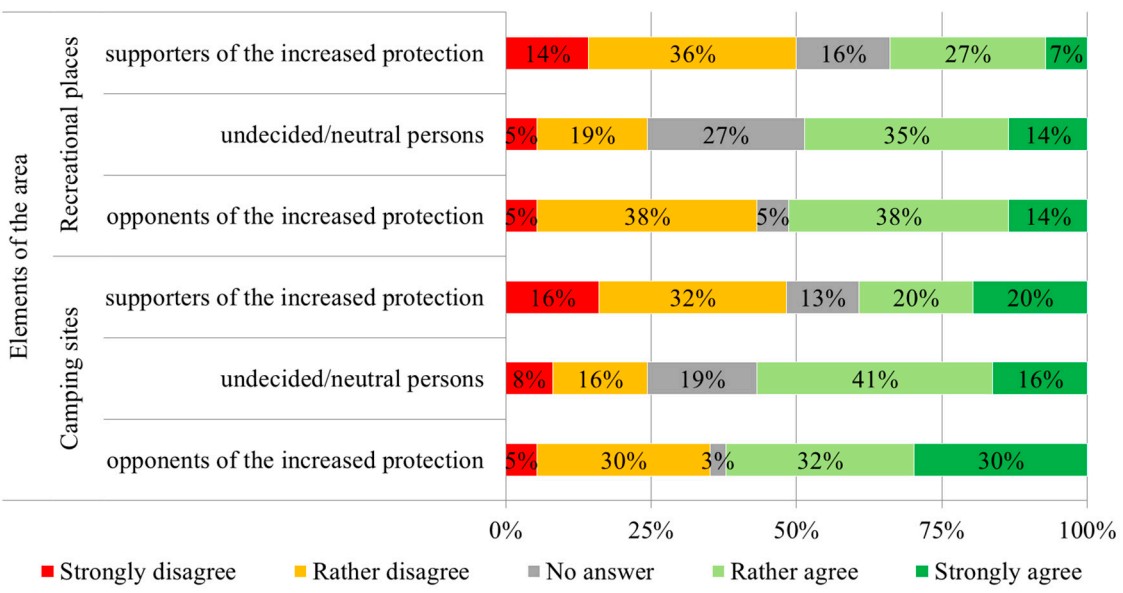

**Figure 7.** Comparison of closely statistically significant differences in opinions on the functioning of the recreational infrastructure in the forest, considering a breakdown of the respondents into undecided, supporters and opponents of nature protection.

As shown in Figure 7, together (the sum of responses, "strong" and "rather" agree) 40% of the "supporters of the increased protection" and 62% of the "opponents of the increased protection", agreed to develop new camping sites in the Bieszczady forests.

## 4. Discussion

Forests cover 1/3 of the territory of Poland; free access to the forest (except for the forests in national parks and nature reserves) is guaranteed by law [34]. Eighty percent of the forest area in Poland is managed by the State Forests National Forest Holding. This institution plays a significant role in the development of physical activity, including tourism, recreation and sports, in forest areas. However, as in other European countries, we can observe a number of conflicts resulted from a multifunctional model of forest economy [35]. The result of this conflict is decline in the public trust in the foresters' work, which can be observed lately in Poland. According to the CPOR report [36] in 2017 three time more respondents negatively evaluated the activity of the State Forests than in previous research conducted in 2012 (increase from 10% in 2012 to 31% in 2017). There are not only conflicts between visitors to recreational or tourist areas and natural resources, but also between recreational activities. The continental region within which Poland is located faces a different set of challenges than the rest of Europe (Atlantic, Central, Mediterranean or Nordic Region), which is partly due to the legacy, as Bell et al. [35] rightly notes a different set of challenges, partly as a result of the legacy of the Soviet or communist era, restitution of forest lands to their former owners and rural depopulation. The main conflict area is between recreation and nature conservation, which is connected with the fact that most of the protected areas were established in the second half of the 20th century. Many of them were established in traditional recreation sites.

The public opinion survey, "Attitudes of Polish women and men towards environmental protection" conducted in 2018–2019 for WWF Poland [37] shows that 45 percent of Poles are dissatisfied with the current state policy on environmental protection. A vast majority of Poles (80% in total) believe that new national parks should be established in Poland. It is therefore not surprising that also in the group of tourist guides we surveyed, the percentage of people considering the area covered by protection in the Bieszczady Mountains as still insufficient was high.

At the same time, the guides' opinions on the scope of the Bieszczady forest protection were correlated with their opinion on the main problems related to the recreational use of forests, especially those related to a temporary limited access to the forest that is hunting and protection and silviculture works. PCA analysis of correlation of respondent demographic characteristics and answers on recreation impediment due to protection and silviculture works and hunting shows that the answers of people aged under 25 and over 56 are characterized by the lowest variance. Hence, their answers are consistent within the population (people of this age respond similarly—PCA does not show what answer they give, but the raw data shows that replies are, "no answer"). Presumably, due to their age (young/old) they do not feel fully informed and because of their lack of knowledge avoid answering these questions. Young people have little experience and may not have had the opportunity to get acquainted with the topic, while older people possibly do not wish to express their opinion understanding the contradictory social views on the issue at stake.

Nevertheless, it must be noted that in general, tourist guides indicated hunting as the most important factor limiting the development of recreation and leisure in the forests. Organization of hunting in the forest may involve temporary restrictions of forest use for all tourists. Hunting is also associated with a threat to the safety of casual observers. The sense of responsibility for people who are guided around the area may affect the attitude of tourist guides to hunters. Moreover, in the group of respondents who indicated hunting as an important factor limiting the possibility of recreation in forests, there were much more supporters of expanding the area of protected areas than their opponents or indecisive people. The attitude towards environmental protection may be of key importance here. Hunting has a long history in Poland and is subject to many different regulations. Certainly, hunting tourism is an economic and social force with a significant impact on underdeveloped rural, remote and agriculturally marginal areas of Bieszczady. However, for emotional and ideological reasons, hunting is often excluded as an income generation option [38].

The general public in Poland currently has a mainly negative view of hunting and hunters [39]. It may be a consequence of a lack of knowledge, an emotional attitude towards the environment or

negative behaviors displayed by the hunters themselves. The ethical values referring to how nature is used in the hunting tourism and the attitudes towards the hunters affect the general public's opinion on the development of hunting tourism [40]. Research by Wierzbicka et al. [41] shows that among the residents of villages their approach to hunters is positive, while the residents of cities have a more negative or neutral approach. Majority of the respondents is represented by the residents of the cities; thus, it may explain a high percentage of opinions that hunting is the main problem for the development of the tourism and recreation in the Bieszczady forest. It is also surprising why supporters of the extension of protected area, more often to their opponents or neutral respondents, indicated silviculture and protection-related works carried out in the forest as a factor that hinder tourism and recreation in the Bieszczady forests. Giergiczny et al. [42] research indicate that individuals, on average, do not derive positive utility from seeing 'forestry at work'. Thus, the greater the management intensity, the lower the recreational value. Respondents also dislike the high level of residue that results from thinning and felling. They prefer to visit older stands, ones that are more diverse in terms of tree spacing and variation in tree size, and finally, those with a larger number of tree species. A higher level of environmental knowledge often correlates with preferences for places with a more natural look, which is also confirmed by our research [43–45].

Contemporary tourism is increasingly demanding in terms of the quality and value of the nature landscape. Hence, another significant factors reducing comfort of recreation in the Bieszczady forests and indicated by the respondents were related to the violation of the spatial order and esthetic values. Both trash in the close proximity to the routes and recreational places and clearcutting close to tourist trails are, just after hunting, the most popular factors indicated as impediment to the tourism and recreation in the Bieszczady forests. Littering is one of the most important factors causing discomfort among tourists and visitors in the forest [46]. The outcomes of research conducted by Ribe [47] showed that clearcutting areas, regardless shape of their edge line, are perceived as unattractive elements of a landscape. There is a number of works indicating a significance of esthetic values [48,49]. Breiby and Slåtten [50] highlight that future research on the tourism should include an esthetical dimension of both natural and human-affected environment to have a better understanding of a "landscape of experiences" of tourists in the nature-based tourist places. The beauty of landscape, exploration of esthetical feelings motivate people to perform recreational activity in the forests. High esthetic quality of the forest is traditionally perceived as an external effect of a good forest management [44]. It is worth mentioning that tourist trails and forest paths truly symbolize the forest and create a sort of "guide" of the views that enables visitors to observe changes in the forest landscape. At the same time Becker [51] and Sedlak [52] think that construction of a network of transport facilities (forest pathways, trails, clearcutting roads, etc.) represents one of the biggest human interventions in a forest ecosystem. This works burdens environment and damages the landscape. Occasionally, these damages may be restored, but in most cases this is impossible [53].

These opinions are reflected in our research. A majority of the respondents (67%) believed that the network of tourist and recreational routes does not require an extension; also, the Beskidy mountain guides indicated a necessity of public consultations at the stage of determining tourist and recreational routes in the forests. On one hand, this results from an increasing awareness and engagement of the Polish society in a process of planning green areas and development of multifunctional public spaces, as well as quite heavy public criticism towards the actions undertaken by foresters related to the renovation of road infrastructure in the forests. Reported preferences and expectations for tourist trails and forest recreational routes have been a subject of interests of many scientists [20,54] for a long time. Sever and Verbic [18] determined that, e.g., trail users prefer fresh air and soundscape to visual experiences. On the other hand, Reynolds et al. Ravenscroft [55,56] highlight the significance of visual values of the landscape in terms of increasing preferences of the trail visitors. Among other linear elements of the forest recreational developments, only cycle trails and educational trails were more acceptable by the tourist guides. This may be due to an increasing popularity of that activity in the forest as a form of recreation [57,58]. Cycling offers freedom to its users, which seems to be

a very attractive option, considering the contemporary, globalized world. However, educational trails are provided with interpretation signs which may support guides' professional activities; thus, we can observe a high percentage of respondents who agree with a necessity to extend educational path networks.

Among other elements of the recreational forest development, camping sites and bonfire places were indicated as those most needed. Undoubtedly, this is related to the use of the forest by the guides. Relax on a camping site or lighting a campfire are among the most popular elements of many trips in Bieszczady. Camping sites in Bieszczady are often located within the village borders, while they are not commonly available in the forest. The accessibility of bonfire places is another problem. Their number is still insufficient, they are located mainly close to more popular tourist trails or recreational places. Due to the limited number of bonfire places, there is a risk of violating the law and lighting illegal campfires in the forest. A solution to this situation would be designating a greater number of recreation and bonfire places. However, such solution is approved mainly by guides who are against the extension of spatial forms of nature protection and guides who have no opinion on this matter.

The conducted research has a few limitations. We did not include a representative sample of the Polish mountain and tourist guides, especially of the Beskidy mountain guides. Among various associations of mountain guides, there are not any associations which would include only guides who operate in the Bieszczady mountains. Thus, the information gathered for this research comes from broadly understood guides of the mountain hiking tourism. Moreover, while the educational profile of the tourist guides was not a subject of this research, this element can be crucial to explain their opinions and attitudes in terms of nature protection and tourist and recreation development of the Bieszczady forests. Finally, the research was conducted online. It is possible that personal interview surveys would allow us to gather deeper information about guide opinions on the forester activity in the Bieszczady forests.

## 5. Conclusions

As in other European countries, forests in Poland are traditionally used for wood production and, increasing often, recreational and sport purposes, such as walks, runs or cycling which are performed as an organized physical activity [59,60]. It results from growing social awareness about the influence of physical activities on life quality, health and mental condition, as well as from a need to be fit and improve general wellbeing. In addition, we can observe increasing scientific works indicating that the nature, especially forests, helps to "recharge one's batteries", reduce stress and a process of recovery in a natural forest environment is much more permanent than it would be achieved in city parks or other fresh-air facilities in the city [61–63]. Hence, free access to the forest for the public is crucial, considering health policy. On one hand, recreational infrastructure in the forest needs to satisfy needs and expectations of its visitors; on the other hand—it still needs to protect forest resources.

Various social groups (local community, local authorities, nonprofit organizations, national park, forest authorities, tourists and tourist guides) may have different expectations of these infrastructures. Mountain guides influence significantly on a public reception of the actions performed to develop tourism and recreation in the Bieszczady forests. Their views about the nature protection in this region, issues related to the tourism and recreation development or current needs for tourist and recreation infrastructure seem to be more neutral and balanced. Tourist guides have a broad environmental and cultural knowledge of the region. Thus they share their opinions about desired and long-term directions of the nature protection, as well as they propose ways how to make the tourist and recreational infrastructure available in the Bieszczady forests. All of this render guides a significant party to the discussion, as they can provide crucial regional perspectives on topics that are related to local government decisions about Bieszczady development.

The conducted analyses enable to formulate the following conclusions:

- According to the Beskidy mountain guides, in the Bieszczady forests the main factor reducing the possibility of free, recreational use of the forest in the Bieszczady forests concerns temporarily restricted access to the forest caused by hunting or forest-management works;
- A majority of the interviewed guides were against a further development of the touristic and recreational infrastructure in the Bieszczady forest;
- According to the respondents there is a necessity of further spatial extension of the protected area in Bieszczady;
- Supporters of the extension of spatial forms of nature conservation in Bieszczady constitute the most critical group among the respondents in terms of the development of tourist and recreational infrastructure, especially creating new recreation or bonfire places in the Bieszczady forests.

Considering the above conclusions, the following recommendations can be made towards the state and local governments making political decisions on the Bieszczady forests development:

- Extension of the protected areas in Bieszczady should be seriously considered;
- Tourist infrastructure in the region is adequate—with the notable exception of camping sites and recreational places;
- Disruptive activities such as hunting and forest-management works should be avoided in the vicinity of tourist trails.

The mountain guides represent important parties and should participate in the public debate about this matter.

**Author Contributions:** E.J. collected literature review, synthesis, wrote methodology, discussions and conclusions. E.B. contributed to prepare a partial text of manuscript and interpret the data. J.P. conducted the survey, performed statistical analysis and contributed to editing. All authors contributed to each section of the paper at different degrees. All authors have read and agreed to the published version of the manuscript.

**Funding:** This research received no external funding.

**Conflicts of Interest:** The authors declare no conflict of interest.

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
