# Peer review of "Forest Tourism and Recreation Management in the Polish Bieszczady Mountains in the Opinion of Tourist Guides"

_sustainability, doi:10.3390/su12197967_

Round 1
Reviewer 1 Report
Thank you for your resubmission. I feel that the paper has been much improved by the range of changes that you have made. The introduction is now more coherent and the method better explained. The paper's findings are well structured and presented, if perhaps rather too long. The discussion is now more comprehensive and covers the issues that were previously raised in the first review. I think that the concluding paragraphs could be stronger as they rather fade away at the end rather than ending with a strong, upbeat note.
I think that greater breaking up of the text into paragraphs would aid readability, and the paper is still in need of proofreading and editing, to correct English language errors (although I appreciate that English is not you native language of course).
I wish you the best.
Author Response
Thank you very much for your review. We are grateful that the reviewer appreciated our effort. Taking into account the comments of the second reviewer, we cut the text a little and removed the last figure. As suggested by the reviewer, we have modified the final points in order to improve their importance. We also corrected language errors. Thank you very much for all your comments, they were very helpful and constructive. We can already see a clear difference between the current version of the manuscript and its previous version.
Reviewer 2 Report
I appreciate the researchers' efforts at revising and resubmitting the manuscript. The paper has substantially improved from previous version. The paper is now much improved in the introduction section, research objective, knowledge gap, and the significance of this study is now well defined or structured. New texts in introduction section in page 3 is well written.
Line: 13: Please write correct grammar.
Line: 94: …sensitive to f danger???
Line 213: use full form while using for the first time, PCA and ANOVA
Line 215: I suggest including funding organization in the last section of the paper.
Line 219: …is still too little?? Little in what? Area? Resources? Not clear.
Line 245: I don’t know what this sentences means” …violation of the spatial order and aesthetic values of the space”. Please clarify.
Figure 2 and 3 needs to be explained.
Line 262: how do you distinguished the most significant differences from significant difference?
Line 274: Please write simple sentences. “…the forest much less often than other groups..” what do you mean by this? Write or give the value as well.
Figure 5. I don’t think this figure shows the significant different values. Are all the scales significant different? Please explain more on figure 5 with examples.
Line 282: Please correct grammar.
Line 321: figure 7 shows opponents of the increased protection is strongly agreed by 30%, please explain why? Should it supposed to be less?
Line 327: I don’t understand figure 8. Looking at the Footpaths trend line (red) and labels, it is confusing to make any inference from the chart. Please revise.
I think the discussion has improved from previous version
This paper still need some revision
Author Response
Thank you very much for your constructive and very relevant comments. We took into account all the reviewer's comments. We corrected grammar and letter errors in lines 13, 94 and 282. We provided full descriptions for statistical tests (line 213). We have removed information about financial issues, redrafted the sentences on lines 219 and 245 and described figures 2 and 3 in more detail. Thank you for your comment on line 262, we have improved the text on both lines 262 and 294. We have also developed the description of figures 5 and 7 and removed figure 8. Indeed, the interpretation of figure 8 is not easy, besides, this figure has not necessarily introduced additional thread to our considerations presented in the manuscript.
Once again, thank you very much, we hope that the corrections and additions we made are in line with the reviewer's expectations.
This manuscript is a resubmission of an earlier submission. The following is a list of the peer review reports and author responses from that submission.
Round 1
Reviewer 1 Report
While I applaud the researchers' efforts at understanding the development of tourism and recreation infrastructure, the paper largely describes survey results without really adding much to the knowledge base. Readers of academic journals expect the articles to provide novel ideas on emerging topics or present rigorous empirical analyses on interesting topics. I'm afraid this paper does neither. The dizzying amount of comparative online survey results would likely be better received in a grey literature technical report. The paper needs a much improved and more detailed introduction section and methods section. Research objective, knowledge gap, and the significance of this study are not well defined or structured. The introduction section need to be completely revised. There are no explanation or justification why authors used these variables in the survey question..”nature protection in Bieszczady; factors hindering recreational comfort in the forest (e.g. difficulties caused by temporary restricted access to the forest, difficulties caused by a malfunction of touristic and recreational infrastructure; and violating spatial order and aesthetic values of the space); and need for new elements of tourist and recreational development in the region”. The author needs to back their research questions through scientific literature. There is no explanation for why these questions were asked. Some of the issues such as below must be addressed:
Line 37: what do you mean by other sports?
Line 40-41: add “…physical activity and psychological wellbeing and provide a substantial economic contribution to the local economy (see: Poudel, J., Munn, I. A., & Henderson, J. E. (2017). Economic contributions of wildlife watching recreation expenditures (2006 & 2011) across the US south: An input-output analysis. Journal of outdoor recreation and tourism, 17, 93-99.) and Poudel, J., Henderson, J. E., & Munn, I. A. (2016). Economic contribution of hunting expenditure to the southern United States of America. International Journal of Environmental Studies, 73(2), 236-254.)
Line 52: use full form while using for the first time ( i.a)
Line 141: you don’t put financial support information here. remove.
Line 154: explain why people indicated hunting as the major factor limiting recreation activities…on the contrary, hunting is a recreation activity. How did you define what recreational activities are?
I think the discussion (or lack of it) should be expanded considerably. This section generally just restates the results with very little explanation of the potential reasons for why.
While re-writing the results section, it would be good to make a sub-section of variables and report the results. It is hard to follow what you are talking about.
This paper also needs substantial English editing.
Reviewer 2 Report
My major concern with this paper is that the research aim (as per lines 80-82) does not add anything 'significant' to scholarly knowledge. The discussion is too descriptive and not analytical enough, with inadequate relating of the findings of this research to the extant literature. My own mentor termed this as "so-what-ness" - you have concluded the findings in lines 324-334, but 'so what'? What are the implications of these findings? In lines 298-299 you accept that interviews would have gained deeper information about guides' opinions. I think that a qualitative (or at least mixed-method study) would have helped to attend to these implications, which would have given greater purpose to the paper.
Other specific issues relate to:
- It is a very short paper, probably resulting from the fact that there is no literature review. There is a terse review of the literature in the introduction, but this is inadequate.
- The paragraphs are way too long, which really hinders 'readability'.
- While I accept of course that English is the authors' second language, the use of English (the (in)definite article in particular(eg. lines 17,19,26) is poor and would require serious editing. The grammatical structure of many sentences is also below par (eg. 245-247).
- The introduction has lots of short sentences that are inadequately linked to the previous sentence and read almost like a list.
- There are some sentences that need rewriting for analytical soundness (for example see lines 177-178: surely you expect significant differences between persons who are 'against' and 'for' the spatial extension).